# Indulgent Parenting and the Psychological Well-Being of Adolescents and Their Parents

**DOI:** 10.3390/children10030451

**Published:** 2023-02-25

**Authors:** Qinglan Feng, Ming Cui

**Affiliations:** Department of Human Development and Family Science, Florida State University, Tallahassee, FL 32306, USA

**Keywords:** adolescence, indulgent parenting, parent–adolescent dyads, psychological well-being, APIM

## Abstract

Adolescence is a time for identity development and exploration. Indulgent parenting during adolescence could be developmentally inappropriate and could be associated with adolescent psychological well-being problems. Little research on indulgent parenting, however, has included and investigated both adolescent and parental well-being problems. To extend the literature, the current study used both adolescent and parental reports in a dyadic context to investigate the association between indulgent parenting and the psychological well-being problems of both adolescents and their parents. This study used a sample of 128 adolescent–parent dyads. The findings from the actor–partner interdependence model (APIM) suggested that (1) the adolescent perceptions of behavioral indulgent parenting were significantly related to their own well-being problems; (2) the parents’ perceptions of relational and behavioral indulgent parenting were significantly related to their own well-being problems; and (3) no effects were found between adolescents and their parents. The findings from this study have implications for prevention and intervention programs to improve parenting practices and reduce parents’ well-being problems.

## 1. Introduction

Adolescents face many challenges besides opportunities, such as increasing academic and social pressure and the growing need for independence from their parents, which may contribute to mental health issues, well-being problems and developmental difficulties [1]. Indulgent parenting, characterized as high levels of warmth and low levels of demandingness or behavioral control [2], could become developmentally inappropriate during this stage [3]. Indeed, studies generally suggested negative consequences of indulgent parenting for adolescent development e.g., [4,5,6]. Due to the renegotiation of the adolescent–parent relationship during adolescence and the potential conflict and stress from the practice of indulgent parenting, indulgent parenting could also affect the parents’ own well-being [7,8]. The association between the parents’ indulgence practice and their own well-being, however, was often overlooked.

When adolescents begin to seek autonomy and independence from their parents, the parents may not be aware of such changing demands during this transition period from childhood to adulthood and may continue to expect their adolescent children to depend on and rely on them. Such discrepancy in these beliefs and expectations may lead to differences in the perceptions of indulgent parenting practices by adolescents and their parents, which may affect the adolescents and their parents in distinct ways. It is, therefore, meaningful to investigate indulgent parenting and well-being in the adolescent–parent dyadic context. This study aimed to examine how the adolescent and parental perceptions of indulgent parenting are associated with their own psychological well-being problems (e.g., adolescent perceptions of indulgent parenting related to their own psychological well-being problems) and with each other’s psychological well-being problems (e.g., adolescent perceptions of indulgent parenting related to their parents’ psychological well-being problems).

### 1.1. Theoretical Perspectives

The psychosocial development theory [9] and self-determination theory [10] suggest that adolescence is a critical time for an individual to seek self-acceptance, autonomy, competence, relatedness and independence. These challenges require corresponding changes in their social and relational contexts (e.g., the family context and parent–child relationship). As adolescents begin developing a stable self-concept and a sense of self, they seek autonomy and independence. Indulgent parenting at this stage becomes developmentally inappropriate because it may inhibit the growth of the sense of self by depriving adolescents from the chances to put in effort and take responsibility, which leads to well-being problems [10]. Further, compared to their parents’ view, indulgent parenting could be perceived as more inappropriate in the eyes of adolescents for a lack of satisfaction with their developmentally psychological needs [11]. Adolescents may perceive indulgent parenting as their parents not valuing their individual selves and deeming them less capable of handling things themselves, which contributes to frustration, stress and other negative psychological outcomes, especially when their views collide with the views of their parents.

The family systems theory [12,13] suggests that a family unit is a complex social system with members intensely and emotionally connected. The change in one family member’s functioning could lead to changes in other family members’ functioning. To understand the impact of indulgent parenting, the family systems theory points to its effects on not only adolescent children but also their parents. The parenting stress theory [14] proposes that daily hassles relating to parenting could be stressors that impact the parents’ well-being by inducing parenting stress. Indulgent parenting, characterized by being highly attentive to their children’s needs while tolerating their misbehaviors, may produce more parenting stress related to their own well-being problems [15,16]. In addition, the family systems theory suggests that disagreements between adolescents and their parents may lead to family disorganization and a lack of cohesion, thus negatively impacting each family member’s functioning and well-being [17]. 

At this stage, adolescents may demand more autonomy and independence from their parents. Parents may not realize such demands during this transition period from childhood to adulthood and continue to expect their adolescent children to depend on and rely on them. Such cognitive discrepancy could then lead to differences in the perceptions of parenting practices and negative parent–child interactions that not only affect adolescent well-being but also parental well-being. Taken together, these theories provided support for the investigation in this study. Specifically, with both the adolescent and parental perceptions being meaningful, indulgent parenting could be associated with the well-being problems among both adolescents and their parents. Further, the perceptions of indulgent parenting from adolescents and their parents could be related to not only their own well-being, but also the other’s.

### 1.2. Indulgent Parenting as Perceived by Adolescents and Their Parents

Extending the conceptual framework of parenting, more recent research has operationalized indulgent parenting to reflect parental indulgence in three dimensions, namely material indulgence (e.g., giving children excessive material goods), relational indulgence (being overly protective and overly involved) and behavioral indulgence (e.g., having low behavioral expectations, shielding children from behavioral consequences) e.g., [3,18,19,20]. The limited empirical evidence demonstrated the systematic differences between the parents’ and their children’s perceptions of parenting behaviors [21,22].

A meta-analysis included 85 studies examining the congruence of the children’s and their parents’ perceptions of parenting and showed that, for example, children tended to report a higher level of psychological control than their parents [23]. Veldorale-Griffin et al. studied indulgent parenting with a sample of adolescents and their parents [24]. They examined the perceptions of indulgence and reported that adolescents perceived being regulated by fewer rules (higher levels of behavioral indulgence) than their parents perceived. According to these related research findings, parents and children may not always agree on parenting practices, especially in adolescence, a period when adolescents begin to seek independence from their parents. With such disagreements, both perceptions provide distinctive information to the investigation of indulgent parenting. As a result, indulgent parenting should be studied in a dyadic context that contains the perceptions of both adolescents and their parents.

### 1.3. Indulgent Parenting and the Psychological Well-Being Problems of Adolescents

Consistent with the parenting framework [2], studies have shown that indulgent parenting was associated with internalizing difficulties and psychological well-being problems such as anxiety, distress and withdrawal among children at various developmental stages (e.g., childhood [25]; young adulthood [26]). Clarke et al. suggested that being overindulged might result in negative feelings, such as anger, embarrassment and guilt [3]. Empirical research on the association between indulgent parenting and mental health problems among adolescents, however, was limited. One recent study by Poornima et al. reported that indulgent parenting was associated with the diagnosis with OCD and comorbid anxiety among adolescents [27].

Despite the generally negative consequences reported in these studies, the findings were not always consistent. Some studies showed contrary findings. Indulgent parenting was reported to be related to lower stress and higher life satisfaction [28] and higher self-esteem [29]. The paradox warrants further investigation of the association between indulgent parenting and adolescent well-being problems.

### 1.4. Indulgent Parenting and the Psychological Well-Being Problems of Parents

With most studies investigating the association between indulgent parenting and adolescent well-being, few studies examined the association between indulgent parenting and the well-being problems of the parents. Based on limited studies, indulgent parenting was negatively related to the parents’ well-being. Parents who practiced indulgent parenting reported feeling frustrated and anxious [3]. In a qualitative study examining the parents’ perspectives of their indulgent behaviors, the parents reported that indulgent behaviors take a great deal of time and energy and they felt exhausted and stressed [30]. In another qualitative study, indulgent parents expressed feelings of uncertainty, regret or guilt during the reflective dialoguing process [31]. Though not tested directly, these studies suggested that the negative feelings associated with the indulgence practice could subsequently lead to well-being problems among the parents of the adolescents. Cui, Darling et al. found that a young adult retrospective report on the indulgent parenting they experienced during childhood and adolescence was positively associated with parenting stress and the parents’ well-being problems later on [7].

Research on the parents, though limited, supported the theories that indulgent parenting is unfavorable and has more negative aspects for the parents. Featured, in part, by high levels of responsiveness, indulgent parenting could be more overwhelming and stressful than other parenting styles. The findings from these related studies shed light on the possible association between indulgent parenting and the parents’ well-being problems.

### 1.5. Influences between Adolescents and Their Parents

From the systems theory, the adolescent and parental perceptions on indulgent parenting practices could be related to not only their own but also the other’s psychological well-being. Such limitation in the literature is partially due to the use of the single informant method and overlooking the dyadic influences. For example, Veldorale-Griffin et al. suggested that material indulgence was positively associated with lower levels of life satisfaction among adolescents through stress [24]. For parents, behavioral indulgence was related to lower levels of life satisfaction through stress. This study examined reports by parents and adolescents separately but did not further explore the relation of indulgent parenting to well-being from such dyadic perspectives. This study, however, provided preliminary evidence on the potential disagreement between adolescents and their parents on their perceptions of indulgent parenting and its potential relation to well-being among both adolescents and their parents. To address this gap in the literature, the present study adopted the dyadic method and aimed to further investigate how the adolescent and parental perceptions of indulgent parenting relate to not only their own but also the other’s well-being.

### 1.6. The Present Study

The purpose of this study is to examine the associations between the perceptions of indulgent parenting and the well-being problems of both adolescents and their parents in a dyadic context. Based on related theories and research, we hypothesized that the adolescent perceptions of indulgent parenting would be positively associated with their own psychological well-being problems, and similarly, the parental perceptions of indulgent parenting would be positively related to their own well-being problems (H1s, actor effects). We further hypothesized that the adolescent perceptions of indulgent parenting would be positively associated with their parents’ psychological well-being problems, and similarly, the parental perceptions of indulgent parenting would be positively related to adolescents’ well-being problems (H2s, partner effects). Since previous studies suggested differential effects of the three dimensions of indulgent parenting [3], this study examined each dimension of indulgent parenting separately. Finally, previous studies have shown that adolescent gender, race, ethnicity and family structure have associations with indulgent parenting and the adolescent and parental well-being problems e.g., [7]. The covariates in this study included the gender, race and ethnicity of both the adolescents and parents. In addition, the parental reports of their highest education levels, family structures and family annual incomes were considered.

## 2. Methods

### 2.1. Participants and Procedures

This study used data from a project studying indulgent parenting and adolescent development. The adolescent participants were recruited from several high schools in a southern region in the U.S. Both the adolescents and their primary parents (one parent) participated in this study. Parental consent and child assent were obtained. The adolescent participants were asked to take an online survey (adolescent survey) on a variety of topics relating to their perceptions of indulgent parenting and their own well-being problems along with their demographic information. Their parents were also asked to complete an online survey (parent survey) on their indulgent parenting behaviors, well-being problems and demographics. Each participant received $20 for the completion of the questionnaire.

Since this study focused on adolescent–parent dyads, only those with both an adolescent report and parent report were used in this study. The sample for this study included 128 adolescent–parent dyads. The adolescent participants had an average age of 15.22, ranging from 12 to 18 years old. Among the adolescents, 44% were White, with 44% African American and 12% other races. More adolescents were female (61%) and non-Hispanic or Latino (83%). For the parents, over half of the parent participants reported their ages between 40 to 49 (57%). Regarding race, 51% were White, followed by 42% African American and 7% other races. The majority of the parents were mothers (90%) and non-Hispanic or Latino (88%). Most of the parents reported their education level as bachelors (40%) or higher than bachelors (24%). The mode for the family’s annual gross income category was $25,000 to $50,000 and the median family’s annual gross income category was $75,000 to $100,000.

### 2.2. Measures

#### 2.2.1. Adolescent and Parental Perceptions of Indulgent Parenting

Indulgent parenting was assessed by the 30-item indulgent parenting scale with the same measure for both the adolescents and their parents [26]. Both the adolescent and parent participants were asked to rate to what degree they agree or disagree with the 30 statements examining indulgent parenting behaviors from 1 (strongly disagree) to 5 (strongly agree). The three dimensions of indulgent parenting were examined by 10 items each: material indulgence (e.g., “my mother/father gives me all the clothes I want,” α = 0.89 for adolescents; e.g., “I give him/her all the clothes he/she wants,” *α* = 0.72 for parents), relational indulgence (e.g., “my mother tries to make me dependent on her,” *α* = 0.76 for adolescents; e.g., “I try to make him/her dependent on me,” *α* = 0.70 for parents) and behavioral indulgence (e.g., “when I break a rule outside of home, my mother would help me avoid the consequences,” *α* = 0.79 for adolescents; e.g., “when he/she breaks a rule outside of home, I would help him or her avoid the consequences,” *α* = 0.79 for parents). The scores were summed within each dimension, with some statements scored reversely. A higher score reflected a higher level of indulgent parenting.

#### 2.2.2. Adolescent and Parental Well-Being Problems

A latent variable of the well-being problems was constructed with three indicators: anxiety symptoms, stress and low life satisfaction. The same measurements were used for both the adolescents and the parents. The severity of anxiety was measured by a 10-item shortened version of the Beck Anxiety Inventory (BAI) [32]. The participants were asked to report how they were bothered by the 10 common symptoms of anxiety (e.g., feeling hot, unable to relax, etc.) during the past month from 1 (not at all) to 4 (severely–it bothered me a lot). The scores were summed up with a higher score indicating a higher level of anxiety (*α* = 0.89 for adolescents, *α* = 0.91 for parents). Stress levels were assessed by the 12-item Rhode Island Stress and Coping Inventory [33]. The participants were asked each statement about their own life and rated the frequency. A five-point scale ranging from 1 = never to 5 = frequently was used. The first seven items were summed together to indicate the participant’s stress levels in the past month (*α* = 0.89 for adolescents, *α* = 0.89 for parents). A higher score indicated a higher level of stress. The sample items included “I felt there was not enough time to complete my daily tasks” and “I felt I had more stress than usual.” Life satisfaction was measured by the Satisfaction with Life Scale (SWLS) [34]. Five items/statements (e.g., “in most ways, my life is close to my ideal”) were included for the participants to report their degree of agreement. The participants were asked to rate the statements from 1 (strongly disagree) to 7 (strongly agree). The scores for the five items were reversed and summed up, with a higher score indicating a lower satisfaction with life (i.e., low life satisfaction, *α* = 0.87 for adolescents, *α* = 0.89 for parents).

#### 2.2.3. Covariates

For both the adolescents and the parents, gender was coded as 1 = male and 2 = female; race as 1 = White and 2 = non-White and ethnicity as 1 = Hispanic or Latino and 2 = Non-Hispanic or Latino. Due to the many levels and skewed nature, the education levels were combined and dichotomized as 1 = high school graduate or lower and 2 = college/bachelor/post-bachelor’s degree. The family structure was dichotomized as 1= biological parent and 2 = other. The family annual income was assessed by seven categories: 1 = below 25 k, 2 = 25 k to below 50 k, 3 = 50 k to below 75 k, 4 = 75 k to below 100 k, 5 = 100 k to below 150 k, 6 = 150 k to below 250 k and 7 = 250 k and above.

### 2.3. Analytical Strategies

The key variables included in the present study were the adolescent reports and the parental reports of three dimensions of perceived indulgent parenting (i.e., material indulgence, behavioral indulgence and relational indulgence) and three indicators of their well-being problems (i.e., anxiety, stress and low life satisfaction). The descriptive statistics and correlations among the study’s key variables were conducted in the preliminary analyses. The missing data were examined by Little’s MCAR Test in SPSS 27, indicating that data were missing completely at random (*χ*^2^ (52) = 54.07, *p* = 0.40).

The actor–partner interdependence model (APIM) [35] was used to measure the association between indulgent parenting and well-being for adolescents and their parents. Both the actor effects and partner effects were examined in the model. The actor effects examined whether both the adolescent and parental perceptions of indulgent parenting were associated with their own well-being problems. The partner effects (H2s) were also included to explore how one’s perception of indulgent parenting was associated with the well-being problems of the other person. The proposed model was conducted by SPSS Amos 27 using the full information maximum likelihood (FIML) estimation method. In testing H1s and H2s, the covariates (i.e., adolescent gender, race and ethnicity; parent gender, race, ethnicity, highest education level, family structure and family annual income) were included in the preliminary models, but only the covariates with significant effects/paths were included in the final models.

## 3. Results

### 3.1. Preliminary Analyses

Table 1 presents the descriptive information of all the study variables and demographics. It provides an overview of the means, standard deviations and range of the key variables. The correlations among the key variables are reported in Table 2. The correlations between the adolescent reports and the parental reports on material indulgence (*r* = 0.29, *p* < 0.01), relational indulgence (*r* = 0.22, *p* < 0.05) and behavioral indulgence (*r* = 0.24, *p* < 0.01) were all positive and significant, which indicated an adolescent–parent agreement on indulgent parenting to a certain degree. The small-to-moderate sizes of the correlations, however, emphasized some degrees of discrepancy between the adolescent and parental perspectives.

The preliminary results from the correlations suggested some mixed findings. As expected, the adolescent report of behavioral indulgence (*r* = 0.21, *p* < 0.05) was positively correlated to their own reports of low life satisfaction. Contrary to our expectation, their reports of material indulgence (*r* = −0.27, *p* < 0.01) and relational indulgence (*r* = −0.21, *p* < 0.05) were negatively related to their low life satisfaction. The correlations between the adolescent reports of material and relational indulgence and their own anxiety and stress, though in the expected direction, were not significant. The correlations among the parental reports were more consistent with our *expectations*. Specifically, the parental reports of behavioral indulgence were positively related to their stress levels (*r* = 0.25, *p* < 0.01). The parental reports of relational indulgence (*r* = 0.31, *p* < 0.01) and behavioral indulgence (*r* = 0.19, *p* < 0.05) were positively correlated with their low life satisfaction.

### 3.2. Hypotheses Testing

The actor effects (H1s) and partner effects (H2s) were tested in APIM. All the covariates were initially included. The results suggested that the adolescent gender was positively associated with the adolescent well-being problems. Therefore, the adolescent gender was retained in the final models. As indicated earlier, the three dimensions of indulgent parenting were tested separately. Figure 1 summarizes the results of the APIMs for each indulgence dimension. The model for material indulgence showed a good fit to the data: *χ*^2^ (18) = 25.48, *p* = 0.11; *CFI* = 0.94, *TLI* = 0.86, *RMSEA* = 0.06, *p close*= 0.37. None of the actor and partner effects were significant.

The model for relational indulgence also showed an acceptable fit to the data: *χ*^2^ (18) = 29.34, *p* = 0.04; *CFI* = 0.92, *TLI* = 0.80, *RMSEA* = 0.07, *p close*= 0.22. The results showed a significant actor effect between the parental reports of relational indulgent parenting and their own well-being problems (parental actor effect: *β* = 0.27, *p* < 0.05), suggesting that a higher level of relational indulgence was related to a higher level of well-being problems. The actor effect for adolescents was not significant. No significant partner effects were found. To test the dyadic pattern suggested by Kenny and Ledermann [36], the partner–actor ratio k for the parents was estimated by *k* = *p*/*a*. The *k* value was −0.04, which supported the “actor only” pattern for the parents.

Finally, the model for behavioral indulgence showed an overall good fit to the data: *χ*^2^ (18) = 20.76, *p* = 0.29; *CFI* = 0.98, *TLI* = 0.95, *RMSEA* = 0.04, *p close*= 0.62. The results showed significant actor effects for the well-being of both the adolescents and the parents (adolescent actor effect: *β* = 0.22, *p* < 0.05; parental actor effect: *β* = 0.31, *p* < 0.05). No significant partner effects were found to be statistically significant. To test the dyadic pattern suggested by Kenny and Ledermann [36], the partner–actor ratio k was estimated. The *k* values supported the “actor only” pattern for both the adolescents and the parents (*k* = 0.18 for adolescents and *k* = −0.29 for parents).

## 4. Discussion

Adolescence is a developmentally critical period that is full of physical and psychological changes and challenges [1]. These challenges require corresponding changes in their social and relational contexts, especially regarding the family context and parenting. Indulgent parenting at this stage could be developmentally inappropriate and bring negative consequences to adolescents, e.g., [3,5,37]. Further, though less studied, indulgent parenting could also affect the parents’ own well-being [7,30]. Finally, given the differences in the parent–child perceptions, dyadic effects could be at the center of the investigation [21,24].

To address these issues, this study investigated the associations between the perceptions of indulgent parenting and the psychological well-being problems for both adolescents and their parents using APIM. Specifically, this study examined the associations between the adolescent and parental perceptions of indulgent parenting and their own psychological well-being problems (actor effects) and the associations between one’s perceptions of indulgent parenting and the psychological well-being problems of the other (partner effects). Recent research has operationalized indulgent parenting to reflect parental indulgence in three dimensions: material indulgence, relational indulgence and behavioral indulgence, but has shown some differential effects e.g., [3,18,19,20]. This study extended the literature by conducting three separate models based on the three dimensions. Using a dyadic modeling framework (APIM), the study investigated several important psychological well-being problems, including symptoms of anxiety, stress and low life satisfaction.

Based on related theories and literature, we hypothesized that the adolescent perceptions of indulgent parenting would be positively associated with their own psychological well-being problems (H1, adolescent–actor effect). The results from our study suggested that the adolescent perceptions of behavioral indulgence were related to their own psychological well-being problems. We further hypothesized that the parental perceptions of indulgent parenting would be positively associated with their own problems in psychological well-being (H1, parent–actor effect). The results of this study support that the parental perceptions of relational and behavioral indulgence in parenting were positively associated with their own psychological well-being problems. We also hypothesized that one’s own (e.g., adolescent’s) perception of indulgent parenting would be associated with the other’s (e.g., parent’s) well-being problems (H2s, partner effects). The results from this study did not support these proposed partner effects.

### 4.1. Indulgent Parenting and Adolescent Psychological Well-being Problems

According to the psychosocial development theory [9], adolescence is a time when an individual begins to develop a stable self-concept and a sense of self, as well as seeking self-acceptance, autonomy and independence. Parents and adolescents need to renegotiate the developmentally appropriate amount of involvement and control provided by parents to achieve the goodness of fit [38,39]. Too much involvement and too little control in adolescence reflect indulgent parenting, which could negatively impact adolescent psychological well-being in multiple aspects e.g., [4,5,6,28,37].

The results of the current study suggested that the adolescent perceived behavioral indulgence was related to their psychological well-being problems. This finding was consistent with the psychological development theory and self-determination theory [9,10]. When adolescents begin to seek autonomy and independence, behavioral indulgence deprives their chances to take responsibility of their own action, thus related to more anxiety, stress and dissatisfaction with life (e.g., frustration). During adolescence, the parental behavioral indulgence may send adolescents the message that they do not value them as being responsible individuals. Such frustration in psychological needs could be related to a series of psychological problems.

The adolescent perceptions of material and relational indulgence were not found to be related to their psychological well-being. Such findings added to the already mixed findings from previous studies e.g., [28,29]. The preliminary results from the bivariate correlations suggested that material indulgence and relational indulgence were negatively related to low life satisfaction. This may suggest that adolescents who experienced material and relational indulgence may be temporarily more satisfied with their life [28]. However, the results from the APIM did not find significant effects of the perceived material and relational indulgence, either positive or negative. The mixed finding may be due to the cross-sectional nature of the study.

### 4.2. Indulgent Parenting and Parental Well-Being Problems

We found support for relational and behavioral indulgent parenting to be related to parental well-being problems, with higher levels of perceived relational and behavioral indulgence being associated with higher levels of well-being problems among the parents, including anxiety, stress and dissatisfaction with life. This finding was consistent with the parenting stress theory [14] and contributed to the current literature by investigating the implications for the parents [7,30]. Indulgent parenting behaviors, especially relational and behavioral indulgent parenting practices, require a great deal of time and energy, which contribute to extra stress and negative emotions in the parents’ lives, thus impacting parental well-being. With more attention and effort paid to the children’s lives, parents with indulgent parenting styles may neglect to take care of their own well-being. We did not find material indulgence to be related to parental well-being. Combined with the findings from the adolescents, the lack of findings on material indulgence may suggest that material indulgence is less detrimental to both adolescents and their parents. Such findings are consistent with some recent studies e.g., [20].

### 4.3. Influences between Adolescents and Their Parents

The results of our study did not support a significant partner effect and it did not fit with the family systems theory [12,13]. Given the limited research in this area, this finding may suggest that an individual’s well-being is likely to be more influenced by their own perceptions. For adolescents, while they are exploring their identity and being self-focused at this stage, their parents’ views of parenting probably do not matter much to them [9]. For the parents, after parenting their child through childhood, they may be more accustomed to assuming their primary role in the parent–child dyad and are more affected by their own parenting practice than how their children perceive it. It is possible that, over time, the parent–child dynamic may change, and mutual influence would occur [40,41]. The null findings in the partner effects could also be due to the small sample size. Future research could further explore the potential partner effects using longitudinal data.

Finally, the adolescent gender was found to be related to their well-being problems. The results suggested that adolescent girls reported higher levels of well-being problems than adolescent boys. This finding was consistent with the previous research that adolescent girls had a greater vulnerability to psychological difficulties, such as intrafamilial stress and depressive symptoms [42,43].

This study filled a research gap and provided a better understanding of indulgent parenting from adolescent–parent dyads. Our findings called attention to not only adolescent well-being problems but also parental issues [7]. Performing a high level of indulgent behaviors and spending considerable time and energy on their children could become a source of well-being issues among parents. This finding addressed the importance of renegotiating the developmentally appropriate parenting behaviors in adolescence for the well-being of both the adolescents and their parents. Further, research focusing on the indulgent parenting style was scarce. Among the limited studies on indulgent parenting, research rarely treated indulgent parenting as a multi-dimensional construct or examined the potential differences of the dimensions. Our findings suggested that, in general, indulgent parenting was related to negative developmental outcomes among adolescents [3]. However, at least in the short term, certain dimensions of indulgent parenting may not emerge as a negative influence yet for adolescents (material and relational indulgence) and their parents (material indulgence). Combining the dimensions of indulgence parenting may mask the potential differences among the different dimensions. This study provided a new perspective explaining the fixed findings in indulgent parenting.

### 4.4. Limitations and Future Research

The present study had several limitations. First, the sample was relatively small in size, homogeneous in terms of some demographic characteristics and consisted largely of mothers as the participating parents. The moderate sample size may have contributed to the null findings in the association between indulgent parenting and the well-being problems. The one location in the U.S. also limited the generalizability of the findings to other areas in the U.S. and other countries that have different cultural backgrounds. For example, would indulgent parenting in a collectivistic culture have the same implications for adolescents and their parents? Some studies also suggested the importance of the role of a father in parenting e.g., [44,45]. Future studies need to use larger and more diverse samples. Second, the current studies focused on three dimensions of psychological well-being problems (anxiety, stress and low life satisfaction). Other aspects of well-being, such as depressive symptoms, may also be associated with the discrepancies in the adolescent and parental reports and should be investigated [46]. Further, the mechanisms of the associations should be explored. Finally, the data were cross-sectional, which prevented us from investigating the directionality of the associations. Based on the theories and previous research, we proposed and tested a unidirectional model in this study to focus on the associations between the perceptions of indulgent parenting and the adolescent and parental well-being problems. It is possible that the problems in well-being could affect the perceptions of parenting practices [47]. Longitudinal data are needed to examine the potential mutual impacts to further understand the relationships between indulgent parenting and well-being.

### 4.5. Implications

Despite the limitations, the present study extended the limited research by using dyadic analyses to understand the association between indulgent parenting and the psychological well-being of both adolescents and their parents. It is important for adolescents and their parents to renegotiate how much involvement and control parents should exhibit at this particular time of adolescence that are appropriate to the adolescents’ needs, benefitting both the adolescents and their parents. The results of the current study could provide information for practitioners working with adolescents and their families. For example, parent education programs may want to educate parents and adolescents to learn more about the growth of “self” and independence during adolescence. The practitioners could also promote communication and discussions between parents and adolescents about the adjusted needs in adolescence to help them achieve a better mutual understanding and agreement on developmentally appropriate parenting behaviors. More in-time and open communication between parents and adolescents may help develop a more cohesive perspective of involvement [38,39]. In addition, parent education programs could add more content to helping parents adjust and deal with their own stress, anxiety and negative emotions related to indulgent parenting, particularly behavioral and relational indulgence, aimed at improving their own psychological well-being.

## Figures and Tables

**Figure 1 children-10-00451-f001:**
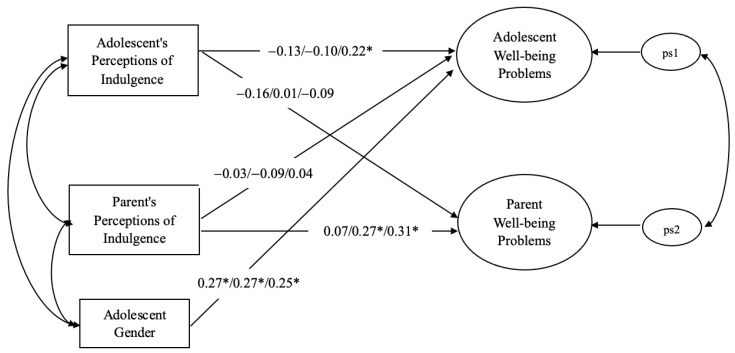
APIM: Indulgence and well-being problems. *β* = material/relational/behavioral indulgence. Adolescent gender was included as a covariate (1 = male 2 = female). * *p* < 0.05.

**Table 1 children-10-00451-t001:** Descriptive and demographic information on the study variables.

Variables	Descriptive Statistics			
	M (%)	S.D.	Min.	Max.
Adolescent Report				
Indulgent Parenting				
Material Indulgence	29.59 ^a^	7.72	10	50
Relational Indulgence	27.74 ^b^	5.93	10	43
Behavioral Indulgence	22.01 ^c^	5.74	12	36
Well-being				
Anxiety	18.21	6.80	10	37
Stress	19.19	6.92	7	35
Low life Satisfaction	17.19	7.11	5	35
Demographics				
Age	15.22	1.12		
Race (White)	44%			
Gender (Female)	61%			
Ethnicity (Non-Hispanic or Latino)	83%			
Parental Report				
Indulgent Parenting				
Material Indulgence	24.20 ^a^	5.03	10	38
Relational Indulgence	25.51 ^b^	4.93	14	42
Behavioral Indulgence	19.71 ^c^	4.93	10	31
Well-being				
Anxiety	14.73	5.57	10	38
Stress	18.86	5.83	7	35
Low life Satisfaction	16.44	6.59	5	33
Demographics				
Age (between 40–49)	57%			
Race (White)	44%			
Gender (Female)	90%			
Family Structure (Biological or Adoptive Mothers)	88%			
Ethnicity (Non-Hispanic or Latino)	88%			
Education (Bachelors and Higher)	64%			
Family Annual Income (below 100 k)	69%			

*Note. N* = 128 dyads. ^a^, ^b^, ^c^: significant differences between the adolescent and parental reports of indulgent parenting (*p* < 0.01 in paired *t*-test).

**Table 2 children-10-00451-t002:** Correlations for the study variables.

Variables	1	2	3	4	5	6	7	8	9	10	11	12
Adolescent Report												
1. Material Indulgence	1.00											
2. Relational Indulgence	0.55 **	1.00										
3. Behavioral Indulgence	0.15	0.26 **	1.00									
4. Anxiety	−0.05	−0.07	0.18	1.00								
5. Stress	−0.12	−0.10	0.17	0.65 **	1.00							
6. Low life Satisfaction	−0.27 **	−0.21 *	0.21 *	0.33 **	0.28 **	1.00						
Parental Report												
7. Material Indulgence	0.29 **	0.14	0.04	−0.04	−0.07	−0.05	1.00					
8. Relational Indulgence	−0.07	0.22 *	0.11	−0.03	−0.18	0.06	0.31 **	1.00				
9. Behavioral Indulgence	−0.16	−0.04	0.24**	0.03	0.06	0.12	0.06	0.34 **	1.00			
10. Anxiety	−0.11	0.02	−0.12	0.09	0.02	−0.01	−0.01	0.12	0.05	1.00		
11. Stress	−0.12	−0.02	0.01	0.18	0.20 *	0.07	0.00	0.12	0.25 **	0.43 **	1.00	
12. Low life Satisfaction	−0.00	0.17	0.04	0.07	0.06	0.17	0.07	0.31 **	0.19 *	0.29 **	0.34 **	1.00

*Note. N* = 128 dyads. * *p* < 0.05, ** *p* < 0.01 (two-tailed).

## Data Availability

Data used in this study are protected due to privacy.

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
