# Peer review of "Indulgent Parenting and the Psychological Well-Being of Adolescents and Their Parents"

_children, 2023, doi:10.3390/children10030451_

Round 1
Reviewer 1 Report
Thank you for the opportunity to read this very interesting and well written paper. There are only two thoughts I want to offer - both in respect to limitations
1. By using the BAI the research was only examining one realm of mental health and thus may have missed some aspects of psychological well-being.
2. As you correctly note there is a geographical limitation - there is also a cultural limitation in that the work is based upon the Western framing of parenting and family.
Thank you
Author Response
Notes to Reviewer 1
“Thank you for the opportunity to read this very interesting and well written paper. There are only two thoughts I want to offer - both in respect to limitations”
Response: We appreciate Reviewer 1 for the positive feedback.
Concern 1. “By using the BAI the research was only examining one realm of mental health and thus may have missed some aspects of psychological well-being.”
Response 1. Reviewer 1 raised a good point. We measured well-being problems as a latent construct with three indicators – anxiety, stress, and low life satisfaction. BAI was used to assess one of the dimensions - symptoms of anxiety. To address Reviewer 1’s concern, we added a sentence at the beginning of this paragraph of the measures of well-being problems to clarify, “ A latent variable of well-being problems was constructed with three indicators: anxiety symptoms, stress, and low life satisfaction.” We also agree with Reviewer 1 and acknowledge that even with three dimensions of well-being problems, there are still many other aspects that we did not examine in this study, as stated in the limitation of the study, “Second, the current studies focused on three dimensions of psychological well-being problems (anxiety, stress, and low life satisfaction). Other aspects of well-being, such as depressive symptoms, may also be associated with the discrepancies in adolescents’ and their parents’ reports and should be investigated [44].”
Concern 2. “As you correctly note there is a geographical limitation - there is also a cultural limitation in that the work is based upon the Western framing of parenting and family.”
Response 2. Reviewer 1 raised a good point. We added further discussion of this limitation in the revision, “The one location in the U.S. also limits the generalizability of the findings to other areas in the U.S. and other countries that have different cultural backgrounds. For example, would indulgent parenting in collectivistic culture be perceived similarly and have the same implications for adolescents and their parents?”
We thank Reviewer 1 for the thoughtful feedback and valuable suggestions.
Reviewer 2 Report
This is a well-written article that illustrates the importance of active and appropriate parenting. It is a strong reminder that parenting requires time and skills and that parenting skills can be taught.
The literature review nicely points out the duality between developing autonomy from parents with the need for some continued dependence on the part of adolescents--and the importance of negotiation between adolescent and parent. The article benefits from a good description of psychosocial development theory, self-determination theory, and family systems theory. The authors offer a literature review for each of the three dimensions of indulgent parenting they are addressing: Indulgent parenting as perceived by adolescents and their parents; indulgent parenting and psychological well-being problems of adolescents; and indulgent psychological well-being problems of parents. Importantly, the authors acknowledge limitations of the literature, especially due to relying only on single informants and not addressing influences of the dyadic relationship. This holds the key to the importance of this work.
In line #147 "and the overlook of dyadic influences" is better as "and overlooking of dyadic influences."
The Methods section is very strong, including the description of the good-sized sample. Hardly a surprise that mothers are 90% of the parent sample. The measures are clearly explained. The section on analytic strategies was especially useful.
I am not a research methodologist/statistician and hope you have a reviewer with this expertise.
The focus on anxiety, stress, and low life satisfaction is important. That adolescents experienced well-being problems from perceived indulgent parenting is a finding of critical importance. That indulgent parents experienced higher levels of anxiety, stress, and low life satisfaction reflects that indulgent parenting is not benefiting them either.
In line # 400 "customed" should be "accustomed"
The Limitations and Future Research Section is excellent, especially due to explicit ideas for future studies.
The Implications section is excellent because it offers concrete suggestions for how parents can acquire stronger parenting skills: parent education programs and clinicians can be more attuned to ways to promote healthy parenting.
Overall, a very strong contribution!
Author Response
Notes to Reviewer 2
“This is a well-written article that illustrates the importance of active and appropriate parenting. It is a strong reminder that parenting requires time and skills and that parenting skills can be taught.
The literature review nicely points out the duality between developing autonomy from parents with the need for some continued dependence on the part of adolescents--and the importance of negotiation between adolescent and parent. The article benefits from a good description of psychosocial development theory, self-determination theory, and family systems theory. The authors offer a literature review for each of the three dimensions of indulgent parenting they are addressing: Indulgent parenting as perceived by adolescents and their parents; indulgent parenting and psychological well-being problems of adolescents; and indulgent psychological well-being problems of parents. Importantly, the authors acknowledge limitations of the literature, especially due to relying only on single informants and not addressing influences of the dyadic relationship. This holds the key to the importance of this work.
…
The Methods section is very strong, including the description of the good-sized sample. Hardly a surprise that mothers are 90% of the parent sample. The measures are clearly explained. The section on analytic strategies was especially useful.
…
The focus on anxiety, stress, and low life satisfaction is important. That adolescents experienced well-being problems from perceived indulgent parenting is a finding of critical importance.
…
The Limitations and Future Research Section is excellent, especially due to explicit ideas for future studies.
The Implications section is excellent because it offers concrete suggestions for how parents can acquire stronger parenting skills: parent education programs and clinicians can be more attuned to ways to promote healthy parenting.
Overall, a very strong contribution!”
Response: We appreciate Reviewer 2 for the positive feedback and strong support of our work.
Concern 1. In line #147 "and the overlook of dyadic influences" is better as "and overlooking of dyadic influences."
Response 1. Changed per Reviewer 2’s suggestion. Thank you.
Concern 2. In line # 400 "customed" should be "accustomed"
Response 2. Corrected. Thank you.
We thank Reviewer 2 for the thoughtful feedback and valuable suggestions.